# PROTEINVISTA: A COMPUTE-EFFICIENT ATOM-LEVEL 3D CNN THAT OUTPERFORMS SEQUENCE TRANSFORMERS IN PROTEIN–LIGAND PREDICTION

## ABSTRACT

Protein function emerges from three-dimensional geometry, but many large-scale prediction pipelines still rely on linear sequence embeddings alone. Although structure-aware protein graph neural networks add residue connectivity, they omit atom-level details and therefore struggle to encode the detailed chemistry of binding sites. Here, we introduce ProteinVista, a compute-efficient 3D convolutional neural network that voxelizes every heavy atom, learns rotation-robust representations through 3D data augmentation, and is pre-trained on over 500 000 AlphaFold-2 structures, which is more than two orders of magnitude less data than used for training state-of-the-art protein language models. Despite its compact size of 123 million parameters, ProteinVista outperforms sequence transformers on three benchmarks that require fine structural resolution: enzyme–substrate classification; transporter–substrate classification; and drug–target inhibition prediction. A simple ensemble with ESM-2 can further improve accuracy, indicating that sequence and structure signals are partly complementary. The results demonstrate that full-atom 3D CNNs are both tractable and superior than protein transformers for structure-dependent tasks. An open-source Python implementation makes ProteinVista easily accessible for application and fine-tuning.

## 1 INTRODUCTION

Protein sequence databases are expanding orders of magnitude faster than experimental annotation can keep pace, widening the knowledge gap between sequence and function (UniProt Consortium, 2021). Scalable *in silico* methods are therefore needed to translate raw sequences into biological insights.

Today's leading approaches encode sequences with transformer-based protein language models (PLMs) such as ESM-2 and ProtT5 (Lin et al., 2023; Elnaggar et al., 2022). Trained on hundreds of millions of sequences, PLMs capture evolutionarily conserved motifs and long-range dependencies that underpin function. Function, however, is ultimately determined by the three-dimensional protein geometry. While large PLMs can infer structure implicitly, as the success of AlphaFold-2 (Jumper et al., 2021), RosettaFold (Baek et al., 2021), and ESMFold (Lin et al., 2023) shows, many applications demand atom-level accuracy. Supplying models with explicit coordinates should therefore benefit tasks that require precise active- or binding-site geometries.

Recent work has begun to feed deep learning models with explicit structural data. Approaches such as DeepFRI (Gligorijević et al., 2021), GearNet (Zhang et al., 2023a), and ESM-GearNet (Zhang et al., 2023b) represent proteins as graphs, where residues are treated as nodes encoded by PLM embeddings and edges capture covalent bonds or spatial proximity. The result is a protein graph that can be processed by a graph neural network (GNN) (Zhou et al., 2020). These models are typically pre-trained using predicted protein structures from AlphaFold-2 (Jumper et al., 2021).

These protein graphs capture residue connectivity but ignore the precise arrangement of atoms. Bond angles, side-chain conformations, and binding pocket geometries are either oversimplified into edge features or omitted altogether. Because large PLMs already infer many residue–residue contacts from sequence alone, augmenting them with a connectivity graph yields only incremental gains, such as faster convergence and small accuracy improvements. For instance, in the extensive ESM-

GearNet study, most investigated protein graph encodings only slightly outperformed the sequence-only ESM-2 baseline, with some even performing worse (Zhang et al., 2023b).

Finer-grained graphs, such as GPS-Fun (Yuan et al., 2024), move closer to atom-level detail, but still treat proteins as topological networks with no direct mapping to 3D space. For tasks that require information on subtle geometric details, such as binding partner complementarity, electrostatics, and active sites, this abstraction can reduce predictive power. A model that uses the full, continuous 3D coordinates of every atom should be better at extracting information about the physics and chemistry that govern molecular interactions.

Convolutional neural networks (CNNs), originally popularized in computer vision for 2D image classification and object detection, have been adapted to volumetric 3D data (Maturana & Scherer, 2015). Between 2017 and 2020, three-dimensional CNNs (3D CNNs) were popular choices for processing 3D protein representations. Early studies used local or simplified input representations: DeepSite locates binding pockets from local grids (Jiménez et al., 2017), EnzyNet predicts EC numbers from the atomic positions of the protein carbon backbone (Amidi et al., 2018), and VoroCNN converts atomic coordinates into a contact graph with Voronoi diagrams to evaluate how well each region of a structure is folded (Igashov et al., 2021). 3DCNN_MQA extends these ideas and uses full atomic 3D CNNs to assess the quality of predicted protein structures (Derevyanko et al., 2018). All these methods were constrained by the limited GPU memory and the absence of today's large public structure datasets such as AlphaFoldDB (Jumper et al., 2021), resulting in models with simplified or localized representations without large-scale pretraining.

Despite recent hardware and data gains, deep learning models acting on full atomic protein 3D structures remain uncommon. It is believed that applying 3D CNNs to entire, high-resolution protein structures is computationally inefficient due to the sparsity of protein atoms in 3D space and the significant memory requirements (Swenson et al., 2020; Gligorijević et al., 2021). Additionally, proteins lack a canonical orientation, whereas vanilla CNNs are not rotation and translation invariant. Solutions therefore require either invariant architectures or extensive orientation augmentation. Nevertheless, recent evidence shows that 3D CNNs for proteins can add complementary information beyond what sequence-based models learn, demonstrated by successfully predicting masked amino acids from local 3D environments (Kulikova et al., 2023).

Here we introduce ProteinVista, a 3D CNN encoder that processes full-atom protein structures (Figure 1). The model maps each atom to a 3D voxel grid and is pre-trained on $\sim 500\,000$ structures from the AlphaFoldDB (Jumper et al., 2021). We aimed to achieve rotation-invariant predictions through extensive 3D augmentations, while grid sizes adaptive to protein size mitigate memory wastage for smaller proteins. Because the pipeline uses only predicted structures, it can be applied to the vast protein sequence space without requiring experimental data.

ProteinVista surpasses current best methods for tasks that depend on fine structural details, such as transporter and enzyme substrate prediction, as well as drug-target interaction prediction. This performance is achieved while requiring 5 times fewer learnable parameters (123 M vs. 650 M for ESM-2) and more than two orders less pre-training data. Embeddings from ProteinVista capture information complementary to sequence-only models; a simple fusion with ESM-2 can significantly improve performance compared to either model alone. An open-source, easy-to-use Python implementation enables the training and application of ProteinVista.

The contributions of this work can be summarized as follows:

- We introduce the first compute-efficient full-atom 3D CNN that was pretrained on large-scale AlphaFold-2 structures to learn rich geometric representations from voxelized proteins.

- We show that ProteinVista provides complementary information to sequence transformers, resulting in enhanced performance on structure-dependent prediction tasks.

- We provide an open-source Python implementation that is easy to use and extend.

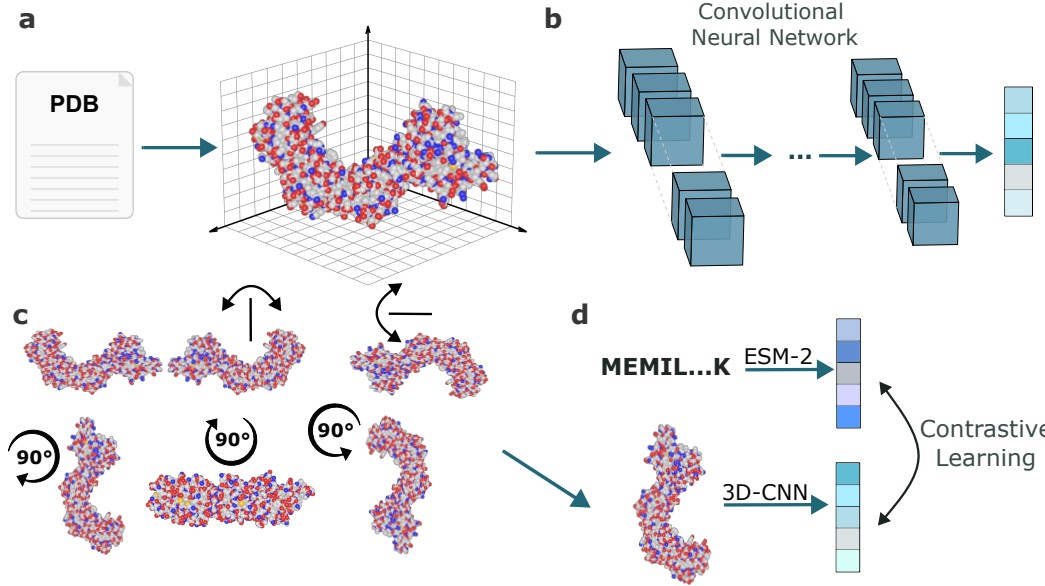

Figure 1: **Overview of the ProteinVista workflow.** **(a)** PDB structures are voxelized into 3D grids that encode the positions of the most common heavy atoms in the protein. **(b)** The resulting grid serves as input to a 3D convolutional neural network (3D CNN). Convolutional features are globally averaged to yield a fixed-length protein embedding. **(c)** To enforce rotational invariance, each structure undergoes random rotations and mirror reflections during training. **(d)** ProteinVista is pre-trained with a contrastive objective that pulls embeddings toward the corresponding ESM-2 embedding of the same protein and pushes them away from those of other proteins.

## 2 PROTEINVISTA ARCHITECTURE AND TRAINING

### 2.1 3D PROTEIN INPUT REPRESENTATIONS

ProteinVista encodes full-atom protein structures as five-channel 3D voxel grids at 1.0 Å resolution (Figure 1). Each channel stores the local density of one of the five most common heavy-atom types: carbon, nitrogen, oxygen, sulfur, and phosphorus. For each voxel these values are stored in a 5-dimensional vector, and hence each protein is represented by a 4D array: three spatial dimensions plus one channel dimension.

Instead of one-hot encoding the presence of atoms at voxels, each atom contributes a continuous density to the 3D grid. For an atom of type $c$ at position $\vec{r} \in \mathbb{R}^3$, its contribution to the $c$-channel of the voxel centered at $\vec{v}$ is $\exp(-\|\vec{v} - \vec{r}\|/\sigma^2)$ with $\sigma = 1$. In contrast to binary encodings, this reduces discretizations artifacts, preserves local geometry, and creates a smoother, more continuous input.

To process proteins of widely differing sizes while limiting empty space in the input boxes, we embed each structure in the smallest of four cubic boxes ($64^3$, $96^3$, $128^3$, or $160^3$ Å$^3$) that fully encloses all atoms. This adaptive boxing minimizes the number of empty voxels and thus lowers memory and compute requirements. Structures that exceed the $160^3$-voxel grid are cropped at the bounding box.

### 2.2 5 BLOCKS OF 3D CONVOLUTIONAL OPERATIONS

ProteinVista consists of a three-dimensional convolutional neural network (3D CNN) that operates directly on the atomic-density voxel grid described above. Each protein input box is processed by five ProteinVista blocks (Table S1), where each block contains one or two 3D convolution layers, a

batch normalization layer, and a ReLU layer, followed by a $2 \times 2 \times 2$ max-pooling operation that halves the spatial resolution. Kernel sizes for the convolutions in layers 1–5 are $7^3$, $5^3$, $3^3$, $3^3$, and $3^3$, respectively. After the final layer, the 3D feature map is reduced in resolution and each voxel is encoded as a 1 024-dimensional vector. Global average pooling over all voxels yields a fixed 1 024-dimensional representation for every protein, independent of the input box size.

For protein-small molecule interaction prediction tasks, this protein vector is concatenated with a 768-dimensional MolFormer small-molecule embedding (Ross et al., 2022). The resulting vector is fed to a two-layer feed-forward network (256 hidden units) with batch normalization, ReLU activation, and dropout to produce the task-specific output prediction. In total, ProteinVista contains $\sim$ 123 M learnable parameters, which is much fewer than the number of parameters of current protein language models that range up to 15 billion (Lin et al., 2023; Hayes et al., 2025).

### 2.3 PRETRAINING

We pre-trained ProteinVista on $> 500\,000$ AlphaFold-2–predicted structures comprising proteins from the Swiss-Prot database (UniProt Consortium, 2021). We explored two pre-training objectives. (i) Multi-task regression: The model was trained to predict 23 *in silico* computed Rosetta scores (Table S2), which summarize per-residue and whole-protein physicochemical properties. Gelman et al. (2025) recently used this objective effectively for pre-training PLMs for mutation-effect prediction. (ii) Contrastive alignment: We projected ESM-2 sequence embeddings (Lin et al., 2023) and ProteinVista structure embeddings into a shared embedding space: the 1 024-dimensional structure embedding produced by ProteinVista is passed through a two-layer projection head to yield a 256-dimensional vector $\mathbf{z}^{\mathrm{PV}}$. A similar projection head maps the ESM-2 embedding to $\mathbf{z}^{\mathrm{ESM}} \in \mathbb{R}^{256}$. For a mini-batch of $n$ proteins we form the similarity matrix $S$ with entries $S_{ij} = \langle \mathbf{z}_i^{\mathrm{PV}}, \mathbf{z}_j^{\mathrm{ESM}} \rangle$ and optimize the symmetric InfoNCE loss with the temperature $\tau = 0.07$ (Radford et al., 2021). The loss encourages paired structure–sequence embeddings from the same protein to be close, and embeddings from different proteins to be distant (Figure 1d).

Ablation experiments (see section 4.2) showed that the contrastive objective yields slightly superior downstream performance (+1.0% relative improvement in $R^2$ on IC50 prediction). Consequently, all task-specific models are initialized with the contrastively pre-trained weights; the task-specific prediction head is replaced with a freshly initialized fully connected neural network, and the entire network is then fine-tuned end-to-end.

### 2.4 DATA AUGMENTATION

Because standard 3D CNNs are orientation-dependent, ProteinVista produces distinct embeddings for rotated copies of the same structure. To this end, every time a protein is presented to ProteinVista, we apply random geometric augmentations during both pre-training and fine-tuning: each input is either left unchanged, mirrored at one of the three Cartesian axes, or rotated by $90°$ around one of those axes, with every option chosen at equal probability (Figure 1c).

Augmenting each protein in a different way each time it is presented to the model enables the network to learn more robust embeddings and predictions, which are less affected by arbitrary rotations of the input protein.

## 3 EXPERIMENTS

In this section, we evaluate the performance of ProteinVista in three protein binding tasks and one protein function prediction task. First, we compare ProteinVista with the widely used ESM-2 models under identical training conditions. Next, we compare ProteinVista's performance against the latest state-of-the-art methods for these tasks.

### 3.1 EXPERIMENTAL SETUP

We compared ProteinVista with two ESM-2 models, one of comparable scale (ESM-2$_{150M}$ with 150 M parameters) and one of larger scale (ESM-2$_{650M}$ with 650 M parameters). For each task, all

models were fine-tuned under identical conditions: we searched for the optimal learning rate, trained the models until no further improvement in performance was observed, and selected the model that performed best on the validation set. ProteinVista and both ESM-2 variants received identical, fixed MolFormer embeddings for the small molecules, and the same two-layer prediction head with 256 hidden units, batch normalization, and ReLU activation function.

This deliberately simple prediction pipeline likely underestimates for all models the peak accuracy achievable with a fully optimized pipeline. Fine-tuning the small-molecule encoder or feeding the fine-tuned protein embeddings as input for separate machine learning models can yield higher scores (Kroll et al., 2023). However, we chose the identical, simple pipeline for all models to isolate the effect of the protein encoder. We also analyzed how ProteinVista's performance changes with more optimized training and prediction pipelines (see sections 3.3 and 4.2).

## 3.2 BINDING PREDICTION TASKS

To quantify the practical benefit of ProteinVista's atom-level information, we benchmarked it against ESM-2 on three problems that require accurate pocket recognition and ligand matching: (i) predicting whether a small molecule is a substrate of a given enzyme (Kroll et al., 2023), (ii) predicting whether a small molecule is a substrate of a given transporter (Kroll et al., 2024a), and (iii) prediction of the inhibition constant $IC_{50}$ for drug–target pairs (Gilson et al., 2016). For details on the datasets, see Table S3.

Table 1: Prediction performance of ESM-2 and ProteinVista on the transporter-substrate (TSP) and enzyme-substrate (ESP) test sets. Higher values indicate better performance. Acc: Accuracy; ROC-AUC: Receiver Operating Characteristic – Area Under the Curve; MCC: Mathews' correlation coefficient.

| Model | Transporter-Substrate (TSP) | | | | Enzyme-Substrate (ESP) | | | |
|---|---|---|---|---|---|---|---|---|
| | Acc | ROC-AUC | MCC | Precision | Acc | ROC-AUC | MCC | Precision |
| ESM-2$_{150M}$ | 88.5% | 0.943 | 0.73 | 0.77 | 91.8% | 0.957 | 0.79 | 0.86 |
| ESM-2$_{650M}$ | 89.3% | 0.947 | 0.74 | 0.79 | 91.9% | 0.955 | 0.79 | 0.86 |
| ProteinVista | 90.8% | 0.952 | 0.77 | **0.85** | 91.8% | 0.951 | 0.78 | 0.89 |
| ESM-ProteinVista | **91.5%** | **0.960** | **0.79** | **0.85** | **93.0%** | **0.961** | **0.82** | **0.91** |
| SPOT | 92.4% | 0.961 | 0.80 | 0.88 | – | – | – | – |
| ProSmith-ESP | – | – | – | – | 94.2% | **0.972** | 0.85 | - |
| Fusion_ESP$_{650M}$ | – | – | – | – | 94.2% | 0.965 | 0.85 | 0.94 |
| ESM-ProteinVista$_{OP}$ | **93.2%** | **0.969** | **0.83** | **0.91** | **94.4%** | 0.967 | **0.86** | **0.95** |

On both binary classification benchmarks, the transporter–substrate prediction (TSP) and enzyme–substrate prediction (ESP) tasks, ProteinVista surpasses or equals the sequence-only ESM-2$_{150M}$ and even the larger ESM-2$_{650M}$ (Table 1). Notably, the ESM-2 models were pre-trained on ∼250 million protein sequences, whereas ProteinVista was pre-trained on only ∼0.5 million structures.

Averaging the predictions of ProteinVista and ESM-2 yields an ensemble that improves on both binary benchmarks (Table 1). This ESM–ProteinVista ensemble surpasses each single model across all metrics, indicating complementary strength: while ESM-2 likely captures evolutionary sequence motifs, ProteinVista contributes explicit 3D pocket geometry. For both prediction tasks, we performed a McNemar's test to assess whether there are significant differences in the error rates of the ESM-ProteinVista ensemble and ESM-2$_{650M}$ alone, and the differences were found to be highly significant ($p < 10^{-13}$ for TSP and $p < 10^{-17}$ for ESP).

The gap between the sequence and structure embeddings widens on the $IC_{50}$ regression task: ProteinVista alone already outperforms both ESM-2 variants, and the ESM–ProteinVista ensemble performs worse (Table 2). We performed a one-sided Wilcoxon signed-rank test on paired squared errors to assess whether ProteinVista yields significantly smaller errors than ESM-2$_{650M}$, and the difference in model predictions is indeed highly significant ($p < 10^{-304}$). Accurate affinity pre-

diction for drug-like ligands relies strongly on fine-grained structural detail, leaving little additional information for the sequence model to contribute. Thus, the relative value of sequence and structure information depends on the task: sequence- and homology-based information is relevant for broad classification problems, whereas detailed affinity prediction requires high-resolution structural context.

Table 2: Prediction performance of ESM-2 and ProteinVista on the BindingDB test set for the IC50 prediction task. Arrows show whether higher ($\uparrow$) or lower ($\downarrow$) values indicate better performance. $R^2$: Coefficient of determination; MSE: Mean squared error; MAE: Mean absolute error.

|  | $R^2 \uparrow$ | Pearson r $\uparrow$ | MSE $\downarrow$ | MAE $\downarrow$ |
|---|---|---|---|---|
| ESM-2$_{150M}$ | 0.60 | 0.78 | 0.86 | 0.71 |
| ESM-2$_{650M}$ | 0.61 | 0.78 | 0.86 | 0.70 |
| ProteinVista | **0.69** | **0.83** | **0.67** | **0.62** |
| ESM-ProteinVista | 0.68 | 0.82 | 0.70 | 0.63 |

### 3.3 COMPARISON TO STATE-OF-THE-ART SUBSTRATE PREDICTION MODELS

To test how performance changes under optimized conditions, we optimized the enzyme- and transporter-substrate training and prediction pipelines for ESM-2$_{650M}$ and ProteinVista: During fine-tuning, we updated the small molecule embedding weights jointly with the ProteinVista encoder, then extracted fine-tuned protein and small molecule MolFormer embeddings for all data points. Using these embeddings, we trained a contrastive network for the binary prediction tasks. Finally, we combined the predictions resulting from the contrastive networks with ProteinVista and ESM-2$_{650M}$ embeddings through simple averaging. We refer to the ensemble model resulting from this optimized pipeline (OP) as ESM–ProteinVista$_{OP}$.

ESM–ProteinVista$_{OP}$ surpasses the current best models. On the transporter–substrate benchmark, it attains an accuracy of 93.2% and a Matthews' correlation coefficient (MCC) of 0.83, exceeding the state-of-the-art SPOT model that achieves an accuracy of 92.3% and an MCC of 0.80 (Table 1). For the enzyme–substrate prediction, ProteinVista$_{OP}$ reaches an accuracy of 94.4% and an 0.86 MCC, slightly outperforming the state-of-the-art models ProSmith-ESP (Kroll et al., 2024b) and Fusion_ESP (Du et al., 2025).

These results confirm that, when coupled with an appropriate training pipeline, ProteinVista serves as a strong foundation for tasks requiring detailed 3D pocket recognition. We expect many structure-dependent protein prediction problems to benefit from integrating ProteinVista as a protein encoder.

### 3.4 HOMOLOGY-RELIANT PREDICTION TASKS

Our binding-task results indicate that ProteinVista is effective when the outcome is determined by an explicit, atom-level description of a protein's binding site. To also test its utility on problems believed to be well explained by sequence homology, we evaluated ProteinVista's performance on Gene Ontology (GO) annotation (Dimmer et al., 2012) predictions, which describe molecular functions, biological processes, and cellular components of proteins.

GO inference is often approached by nearest-neighbour searches: transferring annotations from the closest BLAST hit already achieves reasonable results in the CAFA GO Term prediction challenge (Zhou et al., 2019; Altschul et al., 1990). Consequently, a sequence-based transformer with global attention might outperform a structure-aware CNN that relies on local 3D filters.

We fine-tuned ProteinVista and ESM-2$_{650M}$ for predicting molecular-function GO terms and evaluated them on an independent test set. ProteinVista reached an $F_{max}$ of 0.57, below ESM-2$_{650M}$'s 0.62. Averaging their prediction raised performance marginally to 0.63. Thus, when functional inference depends mainly on conserved motifs or overall homology, structure encoders add limited value.

# 4 ANALYSIS AND ABLATION STUDIES

## 4.1 PREDICTION ACCURACY VARIES WITH SEQUENCE AND STRUCTURAL SIMILARITY

To examine under which conditions ProteinVista performs best, we partitioned the transporter–substrate test set according to three criteria (Figure 4.1 a-c). We first binned proteins by their maximum pairwise sequence identity compared to the training proteins (Figure 4.1a). In high-identity bins, ProteinVista outperforms ESM-2. This indicates that the functional impact of a few amino acid substitutions is captured more effectively from 3D structures than from linear sequences. As identity drops, the gap narrows. In the lowest-identity bin, the models perform similarly, but averaging their predictions improves accuracy compared to each model alone.

We repeated the analysis using the maximum pairwise TM-score, a measure of global structural similarity (Zhang & Skolnick, 2005), instead of sequence identity (Figure 4.1b). The pattern is similar: ProteinVista performs best when the test fold is well-represented in the training set, whereas ESM-2 performs better when similar structures are absent from the training set. Across all similarity ranges, the ESM–ProteinVista ensemble outperforms both single models, showing their complementarity.

ProteinVista relies on AlphaFold-2–predicted structures. We examined how prediction accuracy varies with the quality of those structures (Jumper et al., 2021), estimated as the mean per-residue confidence score (pLDDT; Figure 4.1c). Stratifying the test set by pLDDT shows that ProteinVista performs best on high-confidence structures (pLDDT > 90), suggesting that experimental structures could further improve accuracy. On low-confidence models, ProteinVista performs comparably to ESM-2. Most test proteins, however, fall into the high-confidence range (Fig.4.1d), indicating that the observed gains reflect the majority of cases rather than a small subset.

## 4.2 ABLATION STUDIES

We quantified the effect of key architectural and training choices by starting from the reference ProteinVista model used for the $IC_{50}$ regression benchmark and perturbing one component at a time (Figure 4.1e). Reported changes are relative to the reference coefficient of determination $R^2 = 0.69$ achieved with the full model.

During inference, ProteinVista averages predictions from five randomly augmented views of each protein. Reducing the ensemble to a single view lowered $R^2$ by 6.4%, confirming that multiple orientations are essential for robust inference. Expanding to ten instead of five views yields a small 0.9% gain. Strikingly, disabling rotational and mirror augmentations during fine-tuning had virtually no impact (–0.1 %), implying that the network learns a stable representation already during pre-training and can be fine-tuned on unaugmented data.

Replacing the contrastive alignment to ESM-2 embeddings with multi-task regression on 23 Rosetta energy terms decreased $R^2$ by 1.0%. Both objectives therefore produce competitive encoders, with the contrastive objective offering a slight advantage for affinity prediction. Voxel size resolution also influences performance: reducing the resolution from 1.0Å to 1.5Å per voxel reduced $R^2$ by 1.1%. This indicates that atom-level resolution is important but can be modestly relaxed without catastrophic loss.

## 4.3 COMPUTE AND STORAGE COMPARISON

We compared compute, data, and storage requirements for ProteinVista and the ESM-2 models (Figure 4.3). A single forward pass requires on average 415 Giga Floating Point Operations (GFLOPs) for ProteinVista, 520 GFLOPs for ESM-2$_{650M}$, and 140 GFLOPs for ESM-2$_{150M}$ (Figure 4.3b). Despite similar numbers of mathematical operations compared to ESM-2$_{650M}$, ProteinVista processes 1 000 proteins on one A100 GPU in 20 seconds during training, versus 215 seconds for ESM-2$_{150M}$ and 426 seconds for the ESM-2$_{650M}$ (Figure 4.3c). This suggests that computations for the ProteinVista CNN with only five ProteinVista blocks can be parallelized more efficiently than those required for the ESM-2 models that have many more encoder layers. On the transporter-substrate prediction task, ProteinVista achieved its best performance after 52 epochs and ESM-2$_{650M}$ after 47 epochs, indicating that faster inference does not come at the cost of slower convergence.

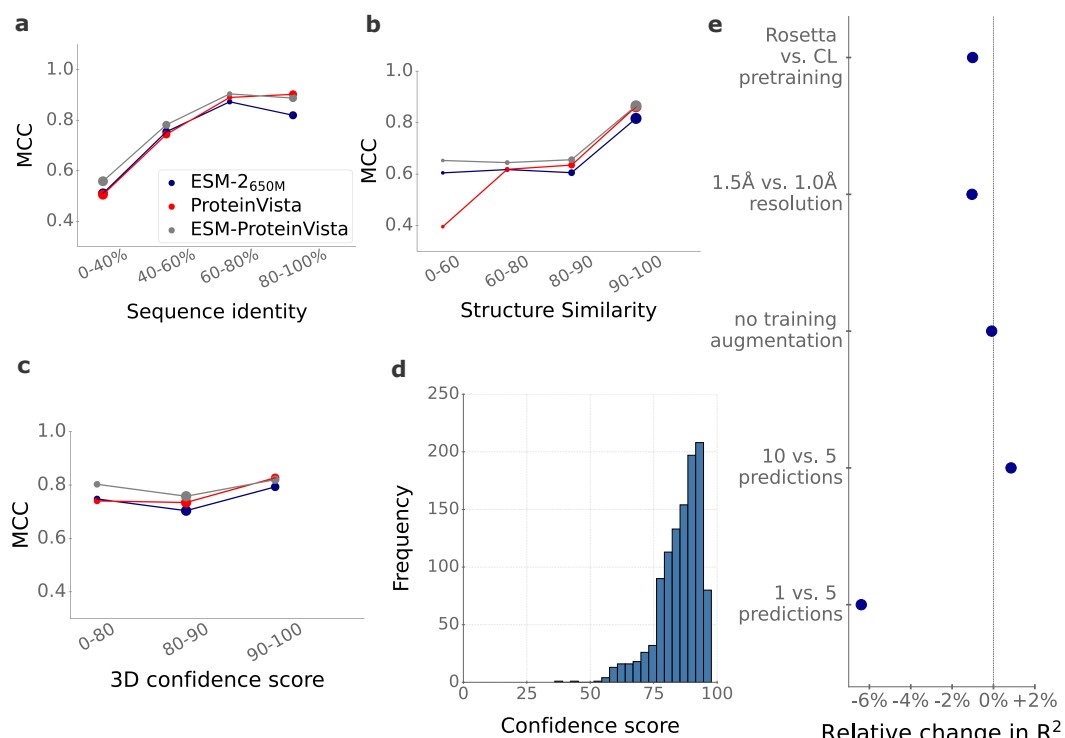

Figure 2: **The ESM–ProteinVista ensemble combines complementary signals from sequence and structure.** Matthews correlation coefficient (MCC) on different subsets of the transporter–substrate test set based on **(a)** maximum sequence identity to the training set and **(b)** maximum TM-score (global structural similarity) to any training structure. **(c)** MCC binned by AlphaFold-2 average pLDDT confidence score of the predicted structures. The areas of the circles in panels **(a)** - **(c)** are proportional to the number of data points in each subset. **(d)** Distribution of pLDDT scores across test proteins. **(e)** Relative change in $IC_{50}$ regression performance for alternative architectural or training choices, expressed with respect to the selected ProteinVista design.

Pre-training requirements diverge even more: ESM-2$_{650M}$ ran for $\sim$ 7 days on 128 H100 GPUs and 250 M different protein sequences, whereas ProteinVista finished pre-training in 48 hours on four A100 GPUs using structures from $\sim$ 0.5 M proteins. The ProteinVista pre-training thus used only about 1% of the GPU-hours. The trade-off is disk space: For example, the 5 800 proteins from the transporter dataset require only 3 MB when storing the linear amino acid sequence (in FASTA format) but $\sim$ 75 GB as float32 coordinate NumPy arrays (Figure 4.3d). Thus, ProteinVista is cheaper in compute and data, but requires larger storage for 3D inputs.

## 5 DISCUSSION AND CONCLUSION

ProteinVista aims to convert the vast amount of available protein structures into detailed biological insights and applies 3D CNNs directly to high-resolution protein structures. By explicitly using atom-level information, the model has a decisive advantage in prediction tasks that require accurate representations of the precise geometry of a protein's structure. Despite being pre-trained on orders of magnitude fewer proteins and computational resources, ProteinVista outperforms or matches the performance of the widely used sequence-based ESM-2 model on three benchmarks of protein-small molecule interaction prediction tasks.

The model achieves this performance without annotations of functionally relevant structural regions, such as binding sites, indicating that it can infer and encode such regions autonomously. Adapting visualization methods such as Grad-CAM (Selvaraju et al., 2017) to 3D convolutions could reveal

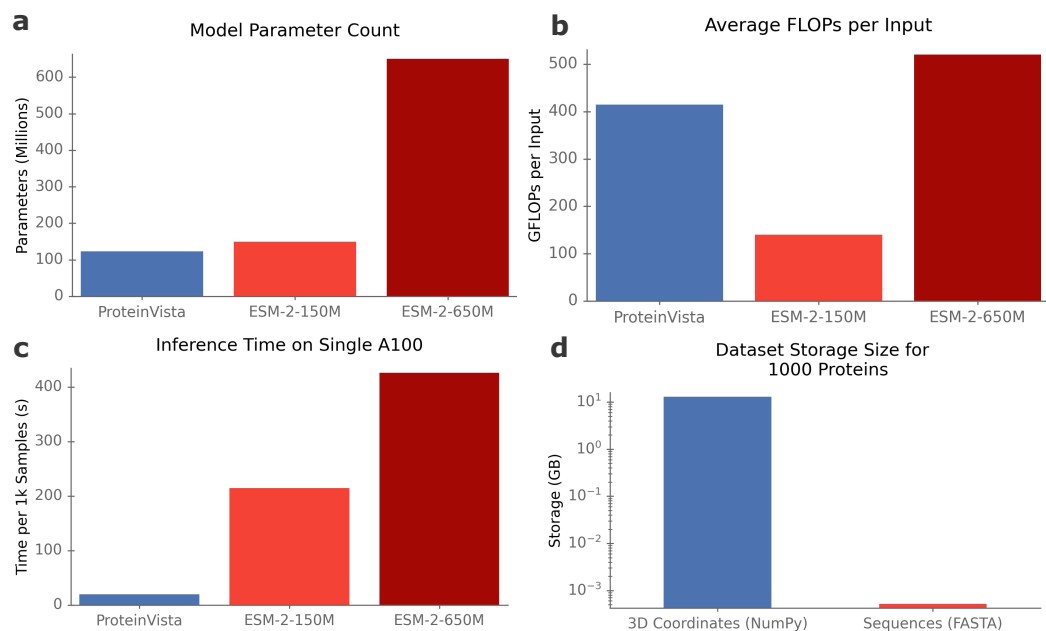

Figure 3: **ProteinVista is compute- and runtime-efficient but storage-intensive.** Comparison of ProteinVista with ESM–2$_{150M}$ and ESM–2$_{650M}$ for **(a)** trainable parameters, **(b)** average GFLOPs per protein, **(c)** training time to process 1 000 proteins (A100 GPU), and **(d)** disk space required to store input data for 1 000 proteins.

the voxels that drive individual predictions. This could pinpoint candidate binding pockets for downstream analysis or experimental validation (Hara et al., 2023).

Effective pre-training strongly influences downstream performance of protein encoders. Here, we compared two pre-training objectives on approximately 500 000 structures: (i) Regression on 33 Rosetta scores, reflecting physicochemical properties and (ii) contrastive learning against ESM-2 embeddings of the linear amino acid sequence. The computer vision literature offers a rich catalog of alternative objectives, such as masked voxel prediction, jigsaw tasks, rotation prediction, and multiview contrastive learning, which could be transferred to volumetric protein data and could yield even more expressive representations. Furthermore, the depth and breadth of ProteinVista's neural network are modest compared to large 2D-CNNs that follow well-established scaling laws (Boopathy & Fiete, 2024). It is not yet known whether similar laws apply in the sparse, high-resolution regime of protein space, but systematically exploring larger, deeper 3D CNNs is a promising path toward further improvements.

A current limitation is that ProteinVista encodes only a single rigid snapshot of each structure. Proteins, however, are dynamic and sample ensembles of conformations that govern function and binding partner affinity. Incorporating alternative snapshots, such as those generated by DynaMine (Cilia et al., 2013), C-Fold (Ellaway et al., 2024), MD-Gen (Jing et al., 2024), AF2$\chi$ (Cagiada et al., 2025), or molecular-dynamics simulations (Hénin et al., 2022), would allow the network to observe how pockets expand or contract. As demonstrated here, even simple augmentations such as random rotations improve performance; dynamic augmentations should supply even richer geometric information and yield more robust predictions.

ProteinVista's strong performance on structure-sensitive tasks suggests several new applications. On the one hand, predicting the functional impact of mutations could profit from the detailed encoding of the geometry of all atoms; accurate mutant structures can be produced by tools such as PDBFixer (Eastman et al., 2017). Furthermore, protein–protein interaction predictions depend on recognizing complementary surface patches across two macromolecules, an inherently structural problem well aligned with ProteinVista's capabilities.

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

# A APPENDIX

## A.1 MODEL ARCHITECTURE

Supplementary Table S1: Layer–by–layer configuration of the ProteinVista model with 5 convolutional blocks.

| # | Layer | In Ch. | Out Ch. | Kernel / Pool | Stride |
|---|---|---|---|---|---|
| 1 | Conv3d | 5 | 128 | $7\times7\times7$ | 2 |
| 2 | BatchNorm3d+ReLU | 128 | 128 | – | – |
| 3 | MaxPool3d | – | – | $2\times2\times2$ | 2 |
| 4 | Conv3d | 128 | 256 | $5\times5\times5$ | 1 |
| 5 | BatchNorm3d+ReLU | 256 | 256 | – | – |
| 6 | Conv3d | 256 | 256 | $5\times5\times5$ | 1 |
| 7 | BatchNorm3d+ReLU | 256 | 256 | – | – |
| 8 | MaxPool3d | – | – | $2\times2\times2$ | 2 |
| 9 | Conv3d | 256 | 512 | $3\times3\times3$ | 1 |
| 10 | BatchNorm3d+ReLU | 512 | 512 | – | – |
| 11 | Conv3d | 512 | 512 | $3\times3\times3$ | 1 |
| 12 | BatchNorm3d+ReLU | 512 | 512 | – | – |
| 13 | MaxPool3d | – | – | $2\times2\times2$ | 2 |
| 14 | Conv3d | 512 | 1024 | $3\times3\times3$ | 1 |
| 15 | BatchNorm3d+ReLU | 1024 | 1024 | – | – |
| 16 | Conv3d | 1024 | 1024 | $3\times3\times3$ | 1 |
| 17 | BatchNorm3d+ReLU | 1024 | 1024 | – | – |
| 18 | MaxPool3d | – | – | $2\times2\times2$ | 2 |
| 19 | Conv3d | 1024 | 1024 | $3\times3\times3$ | 1 |
| 20 | BatchNorm3d+ReLU | 1024 | 1024 | – | – |
| 21 | Conv3d | 1024 | 1024 | $3\times3\times3$ | 1 |
| 22 | BatchNorm3d+ReLU | 1024 | 1024 | – | – |
| 23 | MaxPool3d | – | – | $2\times2\times2$ | 2 |

## A.2 ROSETTA SCORE TERM PRE-TRAINING

We pre-trained ProteinVista as a multi-task regressor that predicts multiple Rosetta energy terms from a single input structure. For every Swiss-Prot protein in the training set we computed three groups of scores with PyRosetta (v3.13): (i) the full-atom score functions `ref2015`, (ii) the low-resolution centroid score functions `score3`, and (iii) four global descriptors from the `ProteinAnalyzer` class: total solvent accessible surface area, number of $C_\alpha$ –$C_\alpha$ contacts, count of hydrophobic residues, and the core-packing metric `two_core_each` (Table S2)

Additionally, if the pairwise Pearson correlation between two Rosetta score terms was greater than 0.95, we randomly selected one of these terms and discarded it, leaving 23 targets per protein. Each target was normalized to a zero mean and unit variance across the dataset. The network was then trained to minimize the mean-squared error averaged over these 23 outputs.

Supplementary Table S2: Rosetta energy terms and global descriptors used as regression targets during pre-training.

| Score terms | |
| --- | --- |
| cenpack | omega |
| contact_all | p_aa_pp |
| dslf_fa13 | pair |
| env | pro_close |
| fa_atr | rama_prepro |
| fa_elec | ref |
| fa_sol | rg |
| hbond_bb_sc | total_hydrophobic |
| hbond_lr_bb | total_sasa |
| hbond_sc | total_score |
| hbond_sr_bb | two_core_each |
| lk_ball_wtd | vdw |

## A.3 CONTRASTIVE PRE-TRAINING

We also pre-trained ProteinVista with a structure–sequence contrastive objective. For every Swiss-Prot entry we computed a fixed 1 280-dimensional sequence embedding with the pre-trained *ESM-2$_{650M}$* model (Lin et al., 2023); these vectors remained frozen during training.

During pre-training, the 1 024-dimensional structure embedding produced by ProteinVista is passed through a two-layer projection head to yield a 256-dimensional vector $\mathbf{z}^{PV}$. A similar projection head maps the ESM-2 embedding to $\mathbf{z}^{ESM} \in \mathbb{R}^{256}$. For a mini-batch of $n$ proteins we form the similarity matrix $S$ with entries $S_{ij} = \langle \mathbf{z}_i^{PV}, \mathbf{z}_j^{ESM} \rangle$ and optimize the symmetric InfoNCE loss with the temperature $\tau = 0.07$ (Radford et al., 2021). The loss encourages paired structure–sequence embeddings from the same protein to be close, and embeddings from different proteins to be distant.

## A.4 DATASETS OVERVIEW

Supplementary Table S3: Dataset sizes for the four benchmark tasks used in this study.

| Dataset | Training size | Validation size | Test size |
| --- | --- | --- | --- |
| Enzyme–Substrate | 733 139 | 5 282 | 13 086 |
| Transporter–Substrate | 23 858 | 2 628 | 6 014 |
| GO Terms (Molecular Function) | 51 514 | 490 | 426 |
| IC$_{50}$ (BindingDB) | 932 442 | 50 000 | 51 720 |

## A.5 SOFTWARE AND CODE AVAILABILITY

All models have been implemented in PyTorch. The source code, pretrained weights and step-by-step training and inference instructions are available as Supplementary Data. All code and data will be made available on GitHub upon publication.

### A.6 Input data preprocessing

We obtained predicted structures in PDB format from the AlphaFoldDB (Jumper et al., 2021). To minimize data processing when running ProteinVista, we converted every raw protein PDB file into a compact NumPy object before model training and validation.

We extract Cartesian coordinates for all atoms and we divide these coordinates by the target resolution (1 Å per voxel). The protein coordinates are then shifted so that all coordinates are non-negative and are at least four voxels away from the grid boundary. This ensures that the convolutional filters can adequately move over the protein surface.

Instead of storing a full 4D tensor for each protein, we save only non-zero entries as 5-tuples $(x, y, z, c, p)$, where $x, y, z$ are integer voxel indices, $c \in \{0, \ldots, 4\}$ is the channel (atom type), and $p > 0$ is the density value. We omit all voxel-channel combinations $(x, y, z, c)$ with values of $p = 0$, resulting in much fewer coordinates than in a fully encoded 4D array. The resulting list is saved as a NumPy `.npy` file and that can be quickly converted into a 4D tensor for model input.

### A.7 Processing small molecules

For prediction tasks that involve a small molecule, every compound is first converted to its canonical SMILES string with explicit stereochemistry. We then use the MoLFormer-XL (Ross et al., 2022), a SMILES Transformer pre-trained on 1.1 billion molecules, to convert the SMILES strings into numerical embeddings: the MoLFormer-XL divides each SMILES string into distinct subparts, the so-called tokens, and computes for each token a numerical representation.

If the small molecule representations are kept fixed, we obtain a single embedding by averaging the token vectors element-wise. If on the other hand we allow to further fine-tune the small molecule embeddings jointly with the protein encoder, the token embeddings are fed into an additional four-layer Transformer encoder, after which they are averaged to obtain the final small molecule vector.

### A.8 ESM-2 model fine-tuning

We fine-tuned the 150 M- and 650 M-parameter versions of ESM-2 (Lin et al., 2023) on every downstream task, using exactly the same train/validation/test splits as for ProteinVista. We applied the same two-layer feed-forward prediction head (256 hidden units, batch normalization, ReLU activation, 10 % dropout, linear output) as for ProteinVista to the updated [CLS] token embedding. For small molecule dependent tasks this embedding is concatenated with the MolFormer small molecule vector (see A.7). All Transformer weights are updated during fine-tuning.

The models were optimized with the AdamW optimizer (weight decay 0.01); we screened learning rates between $1 \times 10^{-6}$ and $1 \times 10^{-4}$ on the validation set and selected the value that maximized the Matthews correlation coefficient (classification tasks) or the coefficient of determination $R^2$ (regression tasks). After training, we selected the model with the best validation performance, and we validated this model on the test set.

### A.9 Evaluation of GO-Term predictions

We evaluated Gene Ontology (GO) term prediction on the Molecular Function subset of the Deep-GraphGO CAFA benchmark (You et al., 2021). The corpus contains 35 092 proteins for training, 490 proteins for validation, and 426 proteins for testing, each annotated with experimentally verified GO terms.

Following the official CAFA protocol we report the $F_{\max}$ statistic, i.e., the maximum harmonic mean of precision and recall obtained when sweeping a threshold $\tau \in [0, 1]$ that binaries the predicted confidence scores $\hat{y} \in [0, 1]$. For a given $\tau$ we convert scores to binary labels $\hat{y}_\tau = \mathbb{1}[\hat{y} \geq \tau]$ and compute precision and recall for every protein. Averaging these quantities over all $N$ proteins yields

$$P(\tau) = \frac{1}{N} \sum_{i=1}^{N} \frac{\text{TP}_i}{\text{TP}_i + \text{FP}_i}, \qquad R(\tau) = \frac{1}{N} \sum_{i=1}^{N} \frac{\text{TP}_i}{\text{TP}_i + \text{FN}_i},$$

where $\text{TP}_i$, $\text{FP}_i$, and $\text{FN}_i$ are the true-positive, false-positive, and false-negative counts for protein $i$. The $F_1$ score at that threshold is

$$F_1(\tau) = \frac{2\, P(\tau)\, R(\tau)}{P(\tau) + R(\tau)},$$

and $F_{\max} = \max_\tau F_1(\tau)$. The metric ranges from 0 (worst) to 1 (best) and is threshold-independent by design.

### A.10 OPTIMIZED PREDICTION PIPELINE

In the optimized ESM-ProteinVista pipeline, we fine-tuned the ESM-2 model and the Protein-Vista models while simultaneously fine-tuning the small molecule embeddings for the protein-small molecule interaction prediction tasks (see A.7). In all cases we extracted fine-tuned representations from the ESM-2 and ProteinVista models. For tasks involving small molecules, we also extracted fine-tuned small molecule embeddings.

For the classification tasks, we trained a separate fully connected neural network with contrastive learning. The predictions from the ESM-2- and ProteinVista-based contrastive models were then averaged, resulting in the finale ESM-ProteinVista$_{\text{OP}}$ predictions.

