# OpenReview forum: "ProteinVista: A compute-efficient atom-level 3D CNN that outperforms sequence transformers in protein–ligand prediction"
_ICLR.cc/2026/Conference — Submitted to ICLR 2026_

### Official Review · Reviewer_TebH · 2025-10-22

**Soundness:** 2
**Presentation:** 3
**Contribution:** 2
**Rating:** 2
**Confidence:** 4

**Summary:**

This paper introduces Protein Vista, a 3D Convolutional Neural Network (CNN) that processes full-atom, voxelized protein structures. The model is pre-trained on over 500,000 AlphaFold-2 structures using a contrastive objective against ESM-2 sequence embeddings. The authors demonstrate that Protein Vista, with only 123M parameters, can outperform the much larger, sequence-based ESM-2 transformer (650M parameters) on several structure-dependent tasks, including enzyme-substrate (ESP) and transporter-substrate (TSP) classification, and drug-target $IC_{50}$ prediction. The paper argues that its full-atom representation is superior to sequence-only models and residue-level graph neural networks (GNNs) for tasks requiring fine geometric detail44.

**Strengths:**

1. Comprehensive experimental evaluation across several biologically meaningful protein–ligand prediction tasks.
2. Compute efficiency: Demonstrates that full-atom 3D CNNs can be trained effectively with modest resources compared with large-scale language model for sequence-based data.
3. Clear complementarity between structure and sequence signals, validated through the ESM–ProteinVista ensemble analysis.

**Weaknesses:**

1. Lack of novelty.
The core idea is about incorporating 3D geometric information in atom-level protein representaion, i.e., voxelizing 3D molecular structures and applying 3D CNN. It is not new. Similar strategies exist in prior works such as VoxMol[1], DeepSite[2], EnzyNet[3], etc. The paper mainly scales this direction with more data and modern augmentation, rather than introducing a fundamentally new modeling principle.
2. Inadequate related work coverage.
The related work section omits many protein-level models that already use 3D geometric information in various ways. Representative examples include VoxMol[1], DeepSite[2], EnzyNet[3], etc.
3. The experiments only compare against sequence-based models (ESM-2, MolFormer). No baselines employing 3D geometric information in protein-level tasks are included, even though these have been widely used in structure-aware prediction. For example, MaSIF[4] for binding-site and ligand interaction prediction. Umol[5] for structure prediction of protein-ligand complexes, etc.

[1]O Pinheiro, Pedro O., et al. "3D molecule generation by denoising voxel grids." Advances in Neural Information Processing Systems 36 (2023): 69077-69097.
[2]Jiménez, José, et al. "DeepSite: protein-binding site predictor using 3D-convolutional neural networks." Bioinformatics 33.19 (2017): 3036-3042.
[3]Amidi, Afshine, et al. "EnzyNet: enzyme classification using 3D convolutional neural networks on spatial representation." PeerJ 6 (2018): e4750.
[4]Gainza, Pablo, et al. "Deciphering interaction fingerprints from protein molecular surfaces using geometric deep learning." Nature methods 17.2 (2020): 184-192.
[5] Bryant, Patrick, et al. "Structure prediction of protein-ligand complexes from sequence information with Umol." Nature Communications 15.1 (2024): 4536.

**Questions:**

1. Why were no structure-based baselines included? And there isn't any section for discussion about related works.
2. The voxel-based method is neither invariant or equivariant for SE(3) group, why do you choose this fashion?

I am willing to modify my score after further revision and discussion.

---

### Official Review · Reviewer_JKWS · 2025-10-31

**Soundness:** 1
**Presentation:** 2
**Contribution:** 1
**Rating:** 0
**Confidence:** 4

**Summary:**

ProteinVista is a new, efficient atom-level 3D convolutional neural network that surpasses two sequence-based transformers in the benchmarks provided by the authors. It encodes protein structures into 3D voxel grids, with a 1A resolution. The model is not invariant to the initial molecular orientations, so the authors augmented the training set by random 90deg rotations and mirror images.

**Strengths:**

This work addresses important protein-related problems.

**Weaknesses:**

- There is no technical novelty in the method
- The authors incorrectly state the problem and do not sufficiently review the state of the art (lack of recent references)
- Point-cloud networks, equivariant networks, convolutions beyond 3D, and pLMs with structural tokens are not discusses
- The work does have technical flaws in the dataset creation and comparisons. Mirror reflection does not make sense for protein structures. There are no mirror- reflecting proteins in nature. 90deg random rotation augmentation is definitely too coarse. No evidence is provided how the computed embeddings differ if one rotates the input protein by an arbitrary amount, e.g. 45 deg
- The experiments present significant data leakage between test and raining sets and must redo the experiments
- The authors must include more recent pLMs with structural tokens as a baseline + compare themselves with dedicated geometrical networks. Provided baselines are outdated and insufficient
- No standard error is provided in the results.The difference in performance between different models may not be significant.
- Data splits for pretraining, fine-tuning and test must be clearly explained with similarity scores provided.
- It is not clear if the multi-task regression to in-silico scores is a good strategy for a foundation model. Additional experiments must be conducted, for example, fine-tuning baselines models with exact values of 23 Rosetta scores, etc. It may happen that Rosetta scores are just sufficient for the presented tasks.

**Questions:**

In the Abstract:
"Although structure-aware protein graph neural networks add residue connectivity, they omit atom-level details and therefore struggle to encode the detailed chemistry of binding sites." - why specifically binding sites? And why do you claim that all graph neural nets omit atom-level details? There are many of these that build graphs starting from atomic positions.

Number of samples in the training set are incorrectly compared with language foundation models. AlphaFold2 and OpenFold were trained on 1-2 orders of magnitude of structures compared to the present work.

"A simple ensemble with ESM-2 can further improve accuracy, indicating that sequence and structure signals are partly complementary." - this claim is a bit controversial, as the network also has access to sequence information and could have learned it.

The motivation is incomplete and somewhat incorrect. The claim about recent graph convolution that only use residue-level representation and are trained only on AF2 models is even beyond the criticism.

"A model that uses the full, continuous 3D coordinates of every atom should be better at extracting information about the physics and chemistry that govern molecular interactions." - please support this claim. Please explain why distance maps cannot be converted into 3D coordinates (what about MDS?).

Point cloud-based networks that have full access to 3D coordinates are completely omitted. Equivariant architectures are not discussed. Nor CNNs beyond 3D, recent works have already introduced invariant and equivariant roto-translational convolutions. Likewise, more recent pLMs that have structure tokens, including ESM3, are not mentioned.

Please provide units to the geometry and the sigma parameter. I suppose it is angstroms. Have the authors tried other sigma values?

90deg random rotation augmentation is definitely too coarse. Please report the difference in the predictions between an arbitrary chosen protein and its copy rotated by 45 deg. Moreover, if you only consider 90deg rotations, you may construct and analytically equivariant model without the need of data augmentation.

For the experiments, you must completely eliminate from the test set samples with TM-score better than 0.5 and sequence identity higher than 30%, otherwise you will have data leakage between training and test sets. You must also compare yourself with more recent pLM models, such as ProST5 and ESM3 that have structural tokens. Also, you need to compare with dedicated geometrical models that are much smaller than what you have, for example PesTO, ScanNet, etc.

---

### Official Review · Reviewer_SEGK · 2025-11-02

**Soundness:** 3
**Presentation:** 3
**Contribution:** 2
**Rating:** 4
**Confidence:** 4

**Summary:**

This paper introduce Protein Vista, a full-atom 3D convolutional neural network (CNN) for protein structure representation and prediction, aiming to outperform sequence-based transformers (like ESM-2) in structure-dependent biochemical tasks. This work represents proteins as voxelized 3D grids of atom densities for key heavy atoms. It utilizes a 5-block 3D CNN architecture with adaptive voxel grids and 3D data augmentation for rotation robustness, pretrained on ~500,000 AlphaFold-2 predicted structures via either (a) multi-task regression on Rosetta energy terms or (b) contrastive alignment with ESM-2 embeddings. It's evaluated on enzyme–substrate prediction, transporter–substrate prediction, IC50 binding affinity regression, and Gene Ontology annotation. It outperforms ESM-2 models of comparable or larger scale on structure-sensitive prediction tasks.

**Strengths:**

1. Demonstration of a compute-efficient full-atom 3D CNN trained at large scale with AlphaFold-2 structures. Clear evidence that explicit 3D atom-level encoding can outperform massive protein language models in certain tasks.
2. Well-structured ablation and performance analyses, including pre-training objectives, voxel resolution, and data augmentation effects. Careful control in experiments: identical small-molecule embeddings and heads across models for fair comparison.
3, Substantial reduction in training compute and data needs compared to transformer-based PLMs. Well-documented trade-off between storage size and compute efficiency.

**Weaknesses:**

1. Limited Benchmark Breadth. Evaluation focuses primarily on protein–ligand binding; broader functional or structural biology tasks (e.g., mutation effects, protein–protein interactions) remain unexplored.
2. Dependence on Predicted Structures. Relies entirely on AlphaFold-2 models, which can be inaccurate for flexible or disordered proteins which introduces systematic bias.
3. Restricted Rotation Invariance. Achieved only via data augmentation, not through inherently equivariant architectures (e.g., SE(3)-equivariant CNNs).
4. While efficient, the voxel-based approach may still scale poorly for very large complexes (cropping may omit structural context).

**Questions:**

Equivariance Alternatives: Did the authors consider or benchmark against SE(3)-equivariant or rotation-invariant CNNs?

---

### Official Review · Reviewer_Jkpa · 2025-11-02

**Soundness:** 2
**Presentation:** 3
**Contribution:** 2
**Rating:** 4
**Confidence:** 4

**Summary:**

The work introduces ProteinVista, an efficient 3D convolutional neural network that operates on voxelized atom-level protein structures. It is pre-trained on ~500k AlphaFold-predicted structures using either Rosetta-based regression or contrastive alignment with ESM-2 embeddings. The model claims to achieve comparable or better performance than sequence-only transformers such as ESM-2 across multiple protein–ligand prediction tasks, including enzyme–substrate, transporter–substrate, and IC50 regression. Despite fewer parameters and significantly less pre-training data, ProteinVista reportedly outperforms larger models, particularly for structure-sensitive tasks, while being faster and more computationally efficient.

**Strengths:**

The paper is clearly written and well organized. The authors present an interesting attempt to revisit full-atom 3D CNNs for protein representation learning. The model’s simplicity and efficiency are appealing, and the experiments cover multiple downstream tasks with fair comparisons to ESM-2. The use of contrastive pre-training with sequence embeddings is sensible and empirically validated. The results show that atom-level representations can provide complementary information to sequence transformers, which is conceptually valuable for integrating structural and sequential knowledge.

**Weaknesses:**

The proposed model architecture and methodology are rather conventional, relying on standard 3D CNN blocks and voxelization. While efficient, the approach does not offer much conceptual novelty beyond scaling previously known techniques to AF structures. The pre-training objectives and augmentations are straightforward, and the reported performance gains are modest or inconsistent across tasks. The study also lacks comparison with baselines besides ESM-2 on certain tasks like IC50 prediction and GO annotation.

**Questions:**

In Table 1, the performance of ESM-ProteinVistaOP is pretty similar to ProSmith-ESP and Fusion ESP 650M, are there analysis to demonstrate the statistical significance?

Also in Table 1, what is the criteria to group the models into two? It is unclear now in the table.

How does the model handle noisy or low-confidence AlphaFold structures, especially in low pLDDT regions?

---

### Official Review · Reviewer_SRme · 2025-11-04

**Soundness:** 3
**Presentation:** 4
**Contribution:** 4
**Rating:** 6
**Confidence:** 3

**Summary:**

The paper presents a simple 3D CNN based model to predict binding affinity and substrate prediction for proteins. The model is pretrained on a large number of (500M) AlphaFold2 structures. Rotational invariance is handle simply with data augmentation. Importantly, the model seems to outperform the state-of-the-art ESM-2 sequence based protein model and -- by some measures -- is more efficient to run as well.

**Strengths:**

- This seems like a very useful tool to solve practical protein structure related tasks
- The architecture is simple and running it seems very reasonable
- It is important to see experimental evidence showing that 3D structure does make a difference
- Interesting to see that the relatively simple data augmentation strategy is sufficient to handle invariance

**Weaknesses:**

- A proper 3D equivariant model might do even better. (They are sometimes also referred to as E(3)-equivariant GNNs)

**Questions:**

Did you try and train a proper E(3) equivariant model on this data too? It is interesting to see that a simple CNN can already improve on ESM-2, but a full E(3) equivariant point cloud based model might do even better. Is there some reason that you did not compare to that?

---

### Meta-Review · Area_Chair_1ric · 2026-01-04

**Summary:**

This submission presents a full-atom voxel-based 3D CNN pretrained on AlphaFold structures and reports strong performance on several protein–ligand interaction tasks when compared with sequence-only transformers. The paper is clearly written, and the empirical observation that explicit 3D geometry can outperform large PLMs on structure-dependent problems is both interesting and practically relevant. The computational efficiency of the approach further adds to its appeal.

However, despite these positive aspects, the core methodological and experimental limitations prevent the paper from reaching ICLR acceptance standards. The approach relies on voxelization and rotation/mirroring augmentations to handle geometric symmetry, but does not provide true SE(3) equivariance or invariance. In contemporary protein structural modeling, geometric symmetry is a fundamental modeling principle rather than an implementation detail, and the absence of strict equivariance weakens the conceptual strength of the method. This issue is compounded by the lack of comparisons against modern equivariant baselines or point-cloud–based geometric models, which currently form the dominant class of structure-aware architectures. Without such baselines, it is difficult to determine whether the reported gains arise from architectural merit, from large-scale pretraining, or from characteristics of the specific tasks.

The related-work discussion also omits several directly relevant voxel- or geometry-based methods, including recent models such as VoxMol, and does not adequately position the contribution within the broader landscape of equivariant and atom-level representations. As a result, the paper tends to overstate its novelty and does not clearly articulate what is fundamentally new beyond scaling a standard voxel 3D CNN to a larger dataset.

In summary, the lack of geometric rigor, the absence of essential baseline comparisons, and the incomplete related-work discussion prevent the submission from providing a sufficiently well-grounded contribution at this time.

I therefore recommend rejecting this paper.

**Reviewer Concerns:**

The authors did not provide a rebuttal.

**Reviewer Scores:**

In the absence of a rebuttal, most reviewers are likely to maintain their initial scores and recommend rejection.

---

### Decision · Program_Chairs · 2026-01-26

Reject